# Reduction in Gingival Bleeding after Atelocollagen Injection in Patients with Hashimoto’s Disease—A Pilot Study

**DOI:** 10.3390/ijerph20042954

**Published:** 2023-02-08

**Authors:** Sylwia Klewin-Steinböck, Marzena Wyganowska

**Affiliations:** Department of Dental Surgery, Periodontal and Oral Mucosa Diseases, Poznan University of Medical Science, 70, Bukowska St., 60-812 Poznań, Poland

**Keywords:** thyroid diseases, atelocollagen, periodontium

## Abstract

Periodontal diseases are one of the main causes of tooth loss and the second most common oral disease after carries. Patients with autoimmune diseases, such as Hashimoto’s disease, are more often vulnerable to infection. In the study group of patients, despite the lack of other signs of gingivitis, bleeding occurred after tooth brushing or minor trauma. Bleeding on probing is the first objective sign of ongoing inflammation. The study was conducted on a group of 17 patients diagnosed with Hashimoto’s disease. The atelocollagen Linerase (100 mg) thinned with 5 mL 0.9% NaCl was used. A total of 0.05 mL of solution was injected into keratinized gingiva, two millimeters above the gingival papillae basement, four times in two-week intervals. The greatest decrease in the number of bleeding points was observed after the first and second injections of atelocollagen. After the third and fourth injections, the average BOP continued to decrease, but the decline was very slow. The use of atelocollagen made it possible to eliminate bleeding symptoms in the study group.

## 1. Introduction

Periodontal diseases are one of the main causes of tooth loss and the second most common oral disease after carries. Worldwide, they are a significant health problem, having an impact on general health. Commonly, they are described as an inflammatory disorder caused and maintained by dental plaque bacteria and their metabolic products. Gingivitis, the most visible sign of periodontal disease is confined to superficial gingival tissue. Clinical signs of gingivitis include edema, erythema, and bleeding following a minor trauma (tooth brushing, sometimes mastication).

Most of the patients with autoimmune diseases, such as Hashimoto’s disease, are more vulnerable to infection. The inflammatory process underlies the etiology of many autoimmune diseases, while the immune system plays an important role in the functioning and maintenance of periodontal stability. Various studies indicate the role of autoimmune diseases in the destruction of periodontal tissue [1,2].

Clinically, healthy gingiva is firm in consistency and strongly attached to the tooth or the underlying alveolar bone. It is pink in color, with no visible swelling, bleeding, or inflammation. Histologically, gingiva is composed of the epithelium and connective tissue separated by a basement membrane. Gingival epithelium covers the underlaying connective tissue and is predominantly cellular in structure. Ultrastructurally, it can be categorized into three different types: oral, sulcular, and junctional epitheliums. The junctional epithelium forms a gingiva attachment to the tooth’s surface, and its structural and functional integrity is crucial to maintain a healthy periodontium [3]. Lying the furthest away, a fibrous connective tissue (lamina propria) is formed by fibroblasts and other cells (i.e., mast cells, macrophages, undifferentiated mesenchymal cells, neutrophils, and plasma cells), fibers, nerve fibers, and blood vessels.

The periodontium is a highly vascular structure with a complex network of blood vessels that supply and drain tissue and has one of the largest end-blood supplies in the body. It has been estimated that 29.1% of all gingival vessels are “invisible” at a histological level and are not functional in healthy gingiva. These dormant capillaries serve as backup vessels when circulation is increased in response to inflammation [4]. Arterial blood is supplied to the periodontal tissues by vascular sources from three different areas—the interdental septa, the periodontal ligament, and the oral mucosa. In the absence of inflammation, the vascular architecture is regular and layered in a network pattern. Gingival connective tissue has two terminal capillary beds. The external network below the marginal gingiva and oral epithelium is formed of terminal hairpin loops extended between epithelial rete pegs. The loops are linked by cross-communication. The internal network is located below the junctional epithelium with capillaries arranged in a flat anastomosing plexus that runs parallel to the enamel from the base of the sulcus to the gingival margin [5]. 

Due to the intensive vascularity, gingival bleeding is one of the first visual signs of periodontal disease and can appear earlier than color change or edema symptoms. From many markers used to describe the disease’s progression, gingival bleeding is objective and an easily assessed inflammation sign. An index based on only one symptom was first used by Mühleman and coworkers. It used grading to describe the severity of bleeding [6]. Bleeding may be induced manually, usually by a periodontal probe. Bleeding on probing (BOP) is an objective symptom of inflammation, shows bleeding simply as present or absent on four surfaces of the tooth, and provides information about the severity of the inflammation [7,8]. BOP scores can be used to differentiate between healthy gingiva and gingivitis [9]. It is widely suggested that patients with numerous bleeding points and deep dental pockets are more prone to further periodontium destruction. 

The destruction of connective tissues starts between 3 and 4 days after plaque accumulation. During the disease process, up to 70% of collagens within the foci of inflammation are destroyed, and the amount and proportion of type V collagen becomes elevated [3]. It was established that the level of collagen loss is the main marker of periodontal disease progression. Collagen, as an essential protein of the connective tissue, through the impact on connective and epithelial tissues accelerates wound healing and tissue regeneration. The external supplementation of collagen is a time-consuming process; however, using collagen as an injection biomaterial supports regeneration in specific areas. These biomaterials can be obtained from animals and humans, or may be synthetized using recombinant genetic engineering techniques. Two different methods are used for the extraction of collagen from animal tissue: pepsin digestion or acid solubilization. Two different collagen forms are obtained: atelocollagen and tropocollagen. Atelocollagen is preferred in commercial use due to the associated cross-species antigenicity of the p-determinant located in the telopeptides [10].

The aim of this study is to evaluate the changes in gingival bleeding (BOP) after equine type I atelocollagen injection was administered to patients with Hashimoto’s disease. 

## 2. Materials and Methods

### 2.1. Materials

The study used the collagen material, Linerase, which is a class III (CE 0477 EPT 0477.MDD.21/4439 EPT 0477.MDD.22/GP0047) medical device in the form of lyophilized type I equine atelocollagen. The equine collagen has the most loosely organized structure and micronization, and because of this it seems to be the most safe. 

### 2.2. Study Design

The study was conducted on a group of 17 patients of different sexes (14 female and 3 male) diagnosed with Hashimoto’s disease. All patients in the study group, despite good oral hygiene and no other symptoms of gingivitis, complained of gingival bleeding after minimal trauma, including tooth brushing. The age of the research group ranged from 32 to 60 years. The study group included three patients with other associated systemic diseases. Two patients were diagnosed with hypertension and one with asthma. The individual measuring of BOP was performed for every patient before the first injection to determine patient’s gingiva condition [Table 1]. The following measurements were obtained before the subsequent injection. In all cases, the atelocollagen (Linerase, 100 mg) thinned with 5 mL 0.9% NaCl was used. A total of 0.05 mL of solution was injected into keratinized gingiva, two millimeters above the gingival papillae basement, four times in two-week intervals (Figure 1). The distance between the points of injections was about 10 mm. During the examination, bleeding was checked on four tooth surfaces. BOP was calculated by dividing the number of bleeding sites by the number of all tested sites and multiplying the obtained value by 100%. The study was conducted with the approval of the bioethics committee, Resolution No 150/17.

## 3. Results

During the first visit, called visit 0 in table and graph, preliminary measurements were performed. During the same visit, the patients were injected with atelocollagen according to the previously described scheme. Every two weeks, new measurements were taken before the subsequent injection, with a total of four injections for every patient. The final measurements were taken two weeks after the fourth injection, during visit 4 (Figure 1). 

The values of the initial measurements in the study group ranged from 75% to 37% (Table 2). 

For each patient, the number of bleeding points was significantly reduced after the first injection of atelocollagen. The BOP value also gradually decreased with each injection (Figure 2).

The greatest decrease in the number of bleeding points was observed after the first and second injections of atelocollagen. After the third and fourth injections, the average BOP (aBOP) continued to decrease; however, the decline was very slow. The clinical condition improved and stabilized. Figure 3 shows the percent decrease in aBOP compared to the initial measurements. The aBOP index decreased by 54% after the first administration of atelocollagen as compared to the initial measurements. Continued treatment resulted in a further decline in aBOP, with a 95% decrease in the aBOP index after the fourth injection as a percentage compared to visit 0.

The most spectacular effect was achieved in the patient with the worst initial situation. After the first two injections, there was a rapid improvement in the clinical condition. The BOP decreased from 75% to 7% (Figure 4). Continued treatment resulted in a further BOP decrease and improvement of the clinical situation, as in other patients.

Out of the entire study group, only one patient showed an increase in BOP after the final injection of atelocollagen due to hospitalization (not related to Hashimoto’s disease) and the total neglect of hygiene procedures. A follow-up examination after 6 weeks showed a slight increase in BOP from 9% to 11%.

During the entire study, two patients had a reduced number of control points due to the need for tooth extraction.

Comparing healthy patients without Hashimoto’s disease with similar oral hygiene habits and a similar clinical condition of gingiva to the study group, two basic differences were noticeable: these patients did not complain of bleeding during brushing and the BOP ranged from 0 to 5%.

## 4. Discussion

Hashimoto’s disease is an autoimmune disorder in which the body’s defenses are directed against its own cells, in this case the thyroid gland. The result is primary hypothyroidism that can cause several health problems. Hashimoto’s thyroiditis has also been considered as one of the causes of periodontal disease. Many studies link periodontal and autoimmune diseases to the chronic inflammation process or oxidative stress [11]. Additionally, in patients with hypothyroidism, other changes in the oral cavity are observed. One of the most common symptoms is xerostomia, which can be indirectly related to gingivitis through the accumulation of plaque. Macroglossia, glossitis, and dysgeusia are also observed in patients with Hashimoto’s disease [12]. Long-term recovery from oral ulcers is also common in these patients. There are also many clinical studies that have shown the co-occurrence of oral lichen planus and Hashimoto’s disease [13]. None of the patients from the study group had the above-mentioned symptoms. 

The concept of staging based on a full-mouth diagnosis presented at the 2017 World Workshop on the Classification of Periodontal and Peri-Implant Diseases and Conditions supports a multidimensional view of periodontitis and the complexity of a patient’s oral rehabilitation needs. According to a new classification, periodontal disease can be divided into three categories: periodontal health and gingival diseases, periodontitis, and other conditions affecting the periodontium. It was agreed that bleeding upon probing is a basic parameter in gingivitis diagnosis. The presence or absence of bleeding is a widely used clinical parameter to determine the presence or absence of periodontal disease progression. 

Inflammation in periodontal tissue occurs in stages. The initial lesion shows no signs of clinical inflammation, but changes can be observed histologically. Junctional epithelium cells produce cytokines and stimulate neutrons to produce neuropeptides, which cause the vasodilatation of local blood vessels and increased vascular permeability. In the subsequent stage, the early lesion, clinical signs of gingival inflammation—bleeding, edema, and the deepening of gingival sulcus—can be observed. The early gingival lesion may persist indefinitely, or it may progress further. If inflammation progresses further, the established lesion advances in stages. Dominant cells within the tissue are macrophages, plasma cells, and T and B lymphocytes. Neutrophils, a major source of matrix metalloproteinase-8 (MMP-8; neutrophil collagenase) and MMP-9 (gelatinase B), release their lysosomal enzymes into the inflamed gingival tissues, causing the destruction of the collagen bundles. Sulcus deepening and the formation of ulcerated epithelium results in bleeding upon probing. The final stage, the advanced lesion, is the transition from gingivitis to periodontitis [14,15]. 

In the initial lesion of gingivitis, the earliest sign of collagen loss can be observed in the perivascular area. The degradation of collagen is induced by bacterial elements, metalloproteinases (MMPs), and resident cells. Much evidence supports the observation that periodontal pathogens stimulate matrix degradation through cytokine production by phagocytes [16]. MMPs are produced and released by resident and inflammatory cells [17]. They are secreted in latent preforms and can be activated by many proteins. It was demonstrated that connective tissue can be a procollagenase store; therefore, collagen destruction can start without the de novo synthesis of metalloproteinases [18]. Plasmin, proteinase derived from plasminogen, has been observed to effectively activate collagenase. The inhibition of bleeding limits plasminogen release, leading to the inhibition of MMP activation.

Histologic and anatomic changes have been observed in gingival microcirculation in inflamed tissue. In healthy gingiva, the vascular network is arranged in a layered, regular pattern [19], with loops bending toward the epithelium. During inflammation, vascular changes begin with vasodilatation, vascular permeability, increased circulation, and the inflow of blood into the area. More capillary loops have been shown, and the changes include an increase in the width and length of the vessels, reduced blood flow [20,21], the vascular plexus presenting an irregular pattern, and microvessels becoming dilatated and looped or twisted [19], leading to gingival bleeding. These changes are essential to the initiation of the inflammation process and to its resolution. 

The histopathology of tissue samples from inflamed gingiva shows dilated blood vessels and collagen fibers and inflammatory cells dispersed in the tissue. In more advanced stages, the disruption of EMC organization and a modified pattern of collagen fibers were observed. The expression of type I collagen in inflamed gingiva was decreased compared to healthy tissue [22]. An in vitro study of fibroblasts derived from diseased gingiva [23] showed no morphological differences compared to fibroblasts from healthy tissue; however, cells from diseased tissue grew slower. Comparing collagen synthesis in healthy and inflamed gingiva, more collagen proteins were found in the medium of healthy fibroblasts. Our previous in vitro study shows that atelocollagen has a direct impact on fibroblasts’ apoptosis. The study shows an increasing number of living cells after 48 and 72 h of incubation under the influence of atelocollagen [24]. 

Collagen is a major protein component of gingiva and plays a dominant role in maintaining the integrity of biological and structural extracellular matrices. It also takes part in tissue repair and wound healing processes. Its primary structure, a triple helix, consists of three α-chains with a repeated structure, Gly-x-y, where x is predominantly proline and hydroxyproline. The presence of glycine in every third position is necessary for the tight packing of the triple helices in the tropocollagen molecule; hydroxyproline contributes to triple-helix stability by forming intramolecular hydrogen bonds [25]. To date, 42 different collagen genes coding for 28 different types of collagens have been identified [26]. Collagen is used in surgery and dentistry for blood clotting, healing, and tissue remodeling; it is a valuable substance in regenerative medicine, a biostimulant, and a filler.

Collagen plays a key role in gingival architecture and periodontal disease. It is a structural component comprising a basement membrane and lamina propria. In healthy gingiva, the average collagen content ranges from 54% to 63% [27]. Main collagen types present in gingiva are I, III, IV, V, and VII. Type I and III collagen are the two major collagenous components of healthy gingiva. Type I accounts for 70 to 95% and type III for 30% of collagen present in gingiva [23]. The ratio between types I and III is 7:1 [26]. Type I is a dominant component of the lamina propria. In healthy gingiva, it is arranged in a parallel fascicular pattern of large, dense bundles of thick fibers. Type III creates a more diffuse, reticular pattern of thin fibers in the lamina propria and basement membrane [22,28]. Type V coats type I and III fibers and has a parallel filamentous pattern. Type IV, a major component of the basement membrane, is found in the lamina densa and within anchoring plaques of the reticular layer [29]. Type IV collagen organizes the three-dimensional meshwork, thanks to the flexibility of the molecule resulting from interrupted Gly-x-y sequences. Type VII, a major component of anchoring fibrils of the basement membrane, forms short, curvy fibrils inserted into the lamina densa and lamina lucida, terminating in connective tissue-anchoring plaques. In the connective tissue, they form loops around collagen fibers, creating a flexible attachment between the basement membrane and underlaying connective tissue. 

Collagens not only serve an important mechanical function within the connective tissue, but they also perform an important function in the cellular microenvironment and are involved in the release and storage of growth factors [30]. Collagen is mainly synthesized by fibroblasts, but may also be secreted by some other cells, such as cementoblasts, odontoblasts, and osteoblasts. In healthy connective tissues, fibroblasts are the cells with low metabolic activity, responding however to signs of tissue damage and bacterial infection. 

Collagen cooperates in the entrapment, storage, and delivery of growth factors. The collagenous matrix binds IGF-I and -II, and collagen I indirectly blocks TGF-β by decorin binding, and with this, collagens create a reservoir of growth factors [31]. Collagen from biomaterials is characterized by its more rapid degradation as compared to endogenous collagen. The injection of atelocollagen can prevent the destruction of tissue collagen by matrix metalloproteinases. Indirectly injected atelocollagen prevents the destruction of the TGF-β reservoir, leading to the activation of the transcription of collagen genes and inhibition of collagenase production [3]. Additionally, degradation products promote the formation of granulation tissue by stimulating the migration and proliferation of fibroblasts, epithelial cells, and vascular endothelial cells [32].

Collagen is considered a non-immunogenic and weak antigen. Its antigenicity is attributed to the interspecies differences in amino acid sequences located in terminal telopeptides. In recent years of periodontal treatment, other methods adopted from aesthetic medicine have been used. One of them is the injection of platelet-rich fibrin or plasma. However, it should be remembered that thyroid diseases have a negative effect on the function of platelets, the functioning of which determines the effectiveness of this treatment [33]. Due to the contraindication of the use of PRPs or PRFs in patients with autoimmune diseases, the use of atelocollagen is safer and not associated with the risk of complications. 

## 5. Conclusions

Due to the frequency of periodontal diseases, the measurement of bleeding tendency should be a standard procedure during an oral examination to help identify the areas at risk of further destruction and to plan proper treatment. At present, periodontal procedures are evolving and surgical interventions are a component of treatment plans. Regenerative procedures strive for the restoration of tissue function and structure. Injectable atelocollagen is used with success in aesthetic medicine to improve skin firmness, thickness, and tension. Based on our clinical trial, we concluded that the use of collagen improves the structure of gingiva, especially in the perivascular area. The results of the study show a positive effect of atelocollagen in patients with Hashimoto’s disease, even though these patients are characterized by delayed tissue healing. The reduction in bleeding was observed two weeks after the first injection and decreased further with subsequent injections. In addition, in patients complaining of severe hypersensitivity, a reduction in hypersensitivity and pain relief was observed due to gingiva thickening and height increase. At the same time, we are aware that atelocollagen cannot replace proper oral hygiene and procedures, such as scaling and root planning. Additionally, further clinical and laboratory studies need to be conducted to understand the mechanism of injectable atelocollagen’s action against gingiva. 

A limitation of this study was the group size. The trial would benefit from a longer study.

## Figures and Tables

**Figure 1 ijerph-20-02954-f001:**
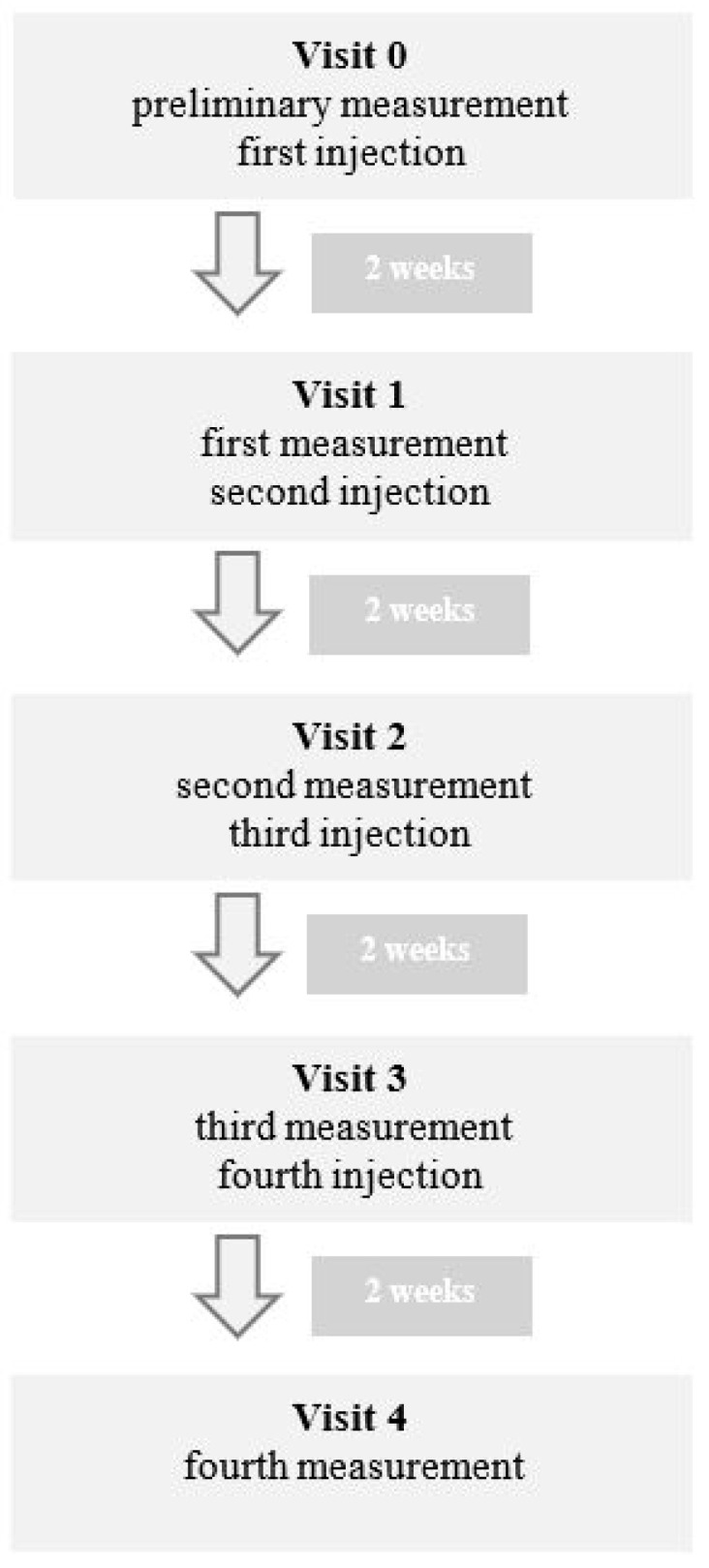
Flowchart of study design.

**Figure 2 ijerph-20-02954-f002:**
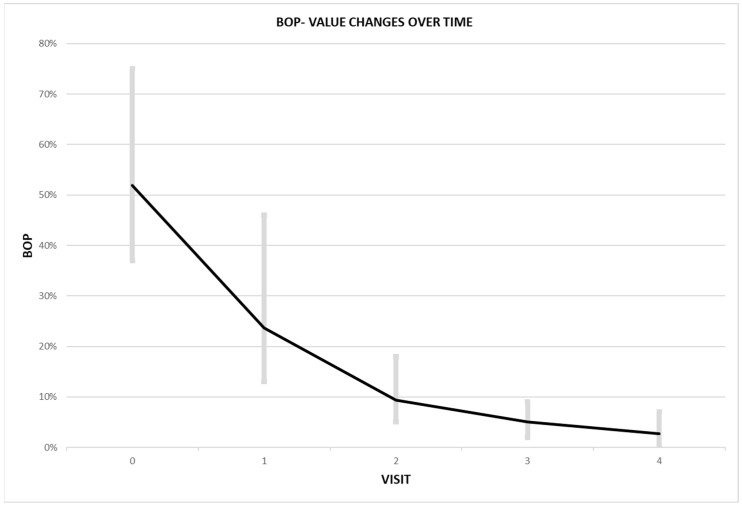
Graph analysis of aBOP changes over time.

**Figure 3 ijerph-20-02954-f003:**
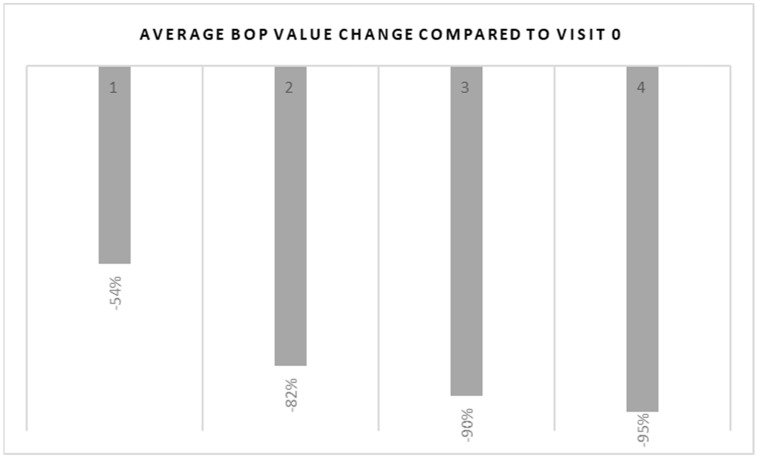
Graphical analysis of BOP decline in comparison to visit 0.

**Figure 4 ijerph-20-02954-f004:**
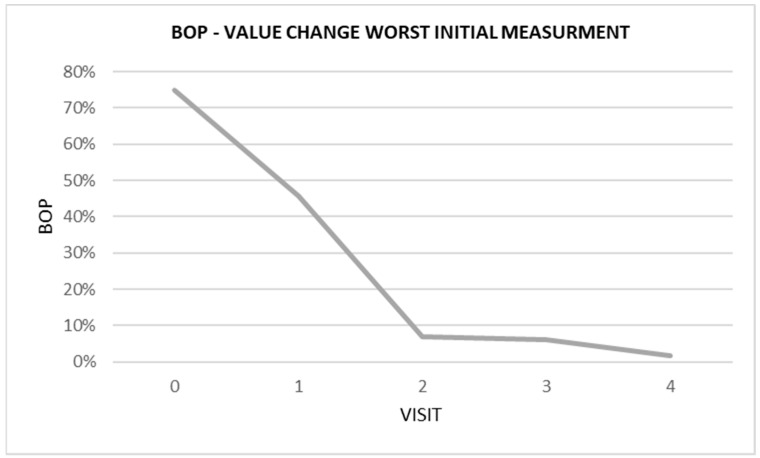
Graphical analysis of BOP changes over time in patient with the worst initial measurement.

**Table 1 ijerph-20-02954-t001:** BOP score describing the condition of the gingiva.

BOP Score	Gingival Status
<10%	healthy gingiva
≥10%–≤30%	localized gingivitis
>30%	generalized gingivitis

**Table 2 ijerph-20-02954-t002:** Percentage results of BOP index after repeated injections.

Visit	0	1	2	3	4
**highest BOP**	75%	46%	18%	9%	7%
**average BOP**	51%	24%	9%	5%	3%
**lowest BOP**	37%	13%	5%	2%	0%

## Data Availability

The data are available on demand.

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
