# Peer review of "Reduction in Gingival Bleeding after Atelocollagen Injection in Patients with Hashimoto’s Disease—A Pilot Study"

_ijerph, 2023, doi:10.3390/ijerph20042954_

Round 1

Reviewer 1 Report

This article adresses an interesting and original topic in specialized literarture,being well structured

I recommend the authors to specify if the patients in the study group have other associated systemic diseases, especially chronic inflammatory diseases

Author Response

Revision Cover Letter

Answers to editor - Manuscript ID: ijerph-2164887

Title: Reduction of gingival bleeding after atelocollagen injection inpatients with Hashimoto's disease - a clinical trial.

Dear Editor,

We thank you and the Reviewers for your thorough review of our manuscript and for the valuable suggestions we have received. The authors carefully analyzed the comments and made every effort to address each one of them. We hope the manuscript after revisions meet your expectation. The authors are grateful for further constructive comments, if any.

Below we provide the point-by-point responses. All modifications in the manuscript are marked in red. We hope that you and the Reviewers find this manuscript acceptable for publication.

Sincerely,

Marzena Wyganowska

Response to Reviewer 1

Comments and Suggestions for Authors

This article addresses an interesting and original topic in specialized literature, being well structured

Response: Thank you very much for review our manuscript and your kind opinion.

I recommend the authors to specify if the patients in the study group have other associated systemic diseases, especially chronic inflammatory diseases

Response: We have completed patient information as follow:

“The study group included three patients with other associated systemic diseases. Two patients were diagnosed with hypertension and one with asthma.”

Reviewer 2 Report

Dear Authors, the paper is interesting but needs a revision.

1. In the introduction there is a lack of information about the material used in the study, its indications and justification for undertaking the study.

2. Flowchart of the study design would be usefull in the M&M chapter.

3. Decription of BOP index is lacking.

4. More information about other oral cavity parameters found in patients with Hashimoto disease is missing.

5. What about the control group / comparative group?

Author Response

Revision Cover Letter

Answers to editor - Manuscript ID: ijerph-2164887

Title: Reduction of gingival bleeding after atelocollagen injection inpatients with Hashimoto's disease - a clinical trial.

Dear Editor,

We thank you and the Reviewers for your thorough review of our manuscript and for the valuable suggestions we have received. The authors carefully analyzed the comments and made every effort to address each one of them. We hope the manuscript after revisions meet your expectation. The authors are grateful for further constructive comments, if any.

Below we provide the point-by-point responses. All modifications in the manuscript are marked in red. We hope that you and the Reviewers find this manuscript acceptable for publication.

Sincerely,

Marzena Wyganowska

Response to Reviewer 2

Comments and Suggestions for Authors

Dear Authors, the paper is interesting but needs a revision.

Response: Thank you very much for review our manuscript and all kind suggestion.

  1. 1. In the introduction there is a lack of information about the material used in the study, its indications and justification for undertaking the study.

Response: We have done our best to add the information suggested. We added the following content into Introduction

“Destruction of connective tissue starts within days 3 and 4 after plaque accumulation. During disease process up to 70% of the collagens within the foci of inflammation is destroyed and the amount and proportion of type V collagen becomes elevated [3]. It was established that level of collagen loss is the main marker of periodontal disease progression. Collagen, as an essential protein of the connective tissue, through the impact on connective and epithelial tissue accelerates wound healing and tissue regeneration. Ex-ternal supplementation of collagen is long time-consuming process but using the collagen as an injection biomaterial supports regeneration in exact point. These biomaterials can be obtained from animals and humans or may be synthetized using recombinant genetic engineering techniques. Two different methods are used for the extraction of collagen from animal tissue: pepsin digestion or acid solubilization. Two different collagen forms are obtaining: atelocollagen and tropocollagen. Atelocollagen is preferred in commercial use due to associated cross-species antigenicity of the p-determinant located in the telopeptides [10].”

Reference added:

  1. Walton, R.S.; Brand, D.D.; Czernuszka, J.T. Influence of telopeptides, fibrils and crosslinking on physicochemical properties of type I collagen films. J Mater Sci Mater Med 2010; 21(2), 451-61.
  2. 2. Flowchart of the study design would be useful in the M&M chapter.

Response: We added the flowchart as suggested in the M&M chapter.

  1. 3. Description of BOP index is lacking.

Response: The information and table 1. “BOP score describing the condition of the gingiva” were added in chapter M&M

BOP score >10% define a patient with healthy gingiva, higher BOP is symptom of local-ized or generalized gingivitis (Tab. 1) [9].

Reference added:

  1. Murakami, S.; Maeley B.L.; Mariotti, A.; Chapple, I.L.C. Dental plaque – induced gingival conditions. J Clin Priodontol 2018, 45(20), 17-27.
  2. 4. More information about other oral cavity parameters found in patients with Hashimoto disease is missing.

Response: We have done our best to add the information suggested. We added the following content into Discussion.

“Additionally, in patients with hypothyroidism, other changes in the oral cavity are ob-served. One of the most common symptoms is xerostomia, which can be indirectly related to gingivitis through the accumulation of plaque. Also observed in patients with Hash-imoto’s disease are macroglossia, glossitis and dysgeusia [12]. Long-term recovery from oral ulcers are also common in these patients. There are also many clinical studies that have shown the co-occurrence of oral lichen planus and Hashimoto’s disease [13]. None of the patients from the study group had the above-mentioned symptoms.”

References added:

  1. Chandna, S.; Bathla, M. Oral manifestations of thyroid disorders and its management. Indian J Endocrinol Metab2011, 15(2), 113-116.
  2. Li, D.; Li, J.; Li, C.; Chen, Q.; Hua, H. The Association of Thyroid Disease and Oral Lichen Planus: A Literature Review and Meta-analysis. Front Endocrinol (Lausanne) 2017, 8, 310.
  3. 5. What about the control group / comparative group?

Response: Healthy patients, without a history of Hashimoto's disease and a similar oral hygiene status, without clinical signs of gingivitis, have a BOP below 5% during a routine examination. Therefore, it is only possible to compare the results of these patients to the preliminary measurements obtained during the visit 0 to the study group. We added the following content into Results.

“Comparing healthy patients without Hashimoto's disease, with similar oral hygiene and similar clinical condition of the gingiva to the study group, two basic differences were noticeable. These patients did not complain of bleeding during brushing and the BOP ranged from 0 to 5%. “

Round 2

Reviewer 2 Report

I didn't see all changes in the main text which were written in the response.

Please add in the title "a pilot" clinicial trial.

Author Response

Dear Reviewer,

we improved the manuscript according your kind suggestions but something happened  during uploading process, which make part of manuscript invisible. We informed about it by email.

We apologize. Hope everything will be good this time.